# Cognitive Remediation in Virtual Environments for Patients with Schizophrenia and Major Depressive Disorder: A Feasibility Study

**DOI:** 10.3390/ijerph18179081

**Published:** 2021-08-28

**Authors:** Adéla Plechatá, Lukáš Hejtmánek, Martina Bednářová, Iveta Fajnerová

**Affiliations:** 1National Institute of Mental Health, 250 67 Klecany, Czech Republic; lukas.hejtmanek@nudz.cz; 2Third Faculty of Medicine, Charles University, 100 00 Prague, Czech Republic; 3Psychotherapeutic Day Center for Psychotic Patients, 733 01 Karviná, Czech Republic; stacionarkarvina-martina@seznam.cz

**Keywords:** schizophrenia, virtual environment, cognitive remediation

## Abstract

Standard approaches to cognitive remediation can suffer from limited skill transferability to patients’ life. Complex virtual environments (VEs) enable us to create ecologically valid remediation scenarios while preserving laboratory conditions. Nevertheless, the feasibility and efficacy of these programs in psychiatric patients are still unknown. Our aim was to compare the feasibility and efficacy of a novel rehabilitation program, designed in complex VEs, with standard paper–pencil treatment in patients with schizophrenia and major depressive disorder. We recruited 35 participants to complete a VE rehabilitation program and standard treatment in a crossover pilot study. Twenty-eight participants completed at least one program, 22 were diagnosed with schizophrenia and 6 with major depressive disorder. Participant’s performance in the representative VE training task significantly improved in terms of maximum achieved difficulty (*p* ≤ 0.001), speed (*p* < 0.001) and efficacy (*p* ≤ 0.001) but not in item performance measure. Neither the standard treatment nor the VE program led to improvement in standardized cognitive measures. Participants perceived both programs as enjoyable and beneficial. The refusal rate was higher in the VE program (8.6%) than in the standard treatment (0%). But in general, the VE program was well-accepted by the psychiatric patients and it required minimal involvement of the clinician due to automatic difficulty level adjustment and performance recording. However, the VE program did not prove to be effective in improving cognitive performance in the standardized measures.

## 1. Introduction

Schizophrenia and major depressive disorders (MDD) are neuropsychiatric conditions representing one of the leading contributing factors to the global burden of disease. Schizophrenia is a disabling and chronic condition, affecting about 1% of the population [1]. MDD has a prevalence of 6% [2]. Both conditions, schizophrenia and MDD, are associated with impairment in cognitive functioning [3,4,5]. The more profound cognitive deficits are reported in patients with psychotic symptoms (e.g., hallucinations, thought disorder, and delusions) [6,7], which are part of schizophrenia’s symptomatology and can be reported in moderate and severe MDD as well [8,9]. The deficit in schizophrenia is generally more profound and broader than in the other psychotic conditions, affecting all cognitive subdomains, with the largest impact on processing speed and verbal memory already manifesting in premorbid states and ultra-high-risk participants [3,10]. Cognitive deficits largely influence patients’ functional outcomes and, in turn, their quality of life [11,12].

Antipsychotic medication is a standard and effective way to treat schizophrenia [13,14,15] but it has only limited impact on cognitive impairment [16,17]. Concerning the effect of antidepressants used for the treatment of MDD, they were shown to have only a small [18] or no effect on cognitive functioning [19,20] with the strongest effect found in selective serotonin reuptake inhibitors (SSRIs) [18].

### 1.1. Cognitive Remediation

Cognitive remediation (CR) represents systematic behavioral interventions focusing on generalizable and durable improvement of cognitive deficits [21]. The interventions can be divided into drill practice, strategy learning approaches, or a combination of both. The drill practice was reported as a beneficial intervention for increasing cognitive performance [22]; strategy learning was argued to lead to larger functional outcome performance [23]. Meta-analyses focusing on the effect of CR show more promising results than the treatment with antipsychotic medication [22,24,25,26]. A large meta-analysis (*n* = 2401) conducted by Wykes et al. [24], pointed to a moderate effect of CR on global cognitive functioning (ES = 0.45), and almost all cognitive domains showed significant improvement. Similar effect sizes, ranging from 0.33 to 0.93, were reported also in MDD [27].

### 1.2. Computerized Cognitive Remediation

The use of computerized systems seems very suitable for the drill practice approach applied in most clinical trials [28]. Computerized cognitive remediation (CCR) enables standardized, precise, and automated stimuli presentation for indefinite time intervals. Moreover, most of the CCR programs can automatically adjust difficulty levels according to the patient’s performance over time. Prikken et al. [22] conducted a large meta-analysis focusing on the efficacy of CCR using drill practice in schizophrenia patients. Prikken’s results are consistent with findings on noncomputerized CR showing moderate effect size for attention (0.31) and working memory (0.38) and smaller effects for processing speed, learning and memory, and verbal fluency; but there was no significant impact on the functional outcome or social cognition, which signals possible limitations of this approach.

### 1.3. Virtual Environments

Standard approaches to CR and neuropsychological assessment are sometimes criticized for being detached from everyday patients functioning and focusing only on the basic cognitive functions [29,30]. In cognitive training, higher ecological validity should result in an easier transfer of practiced abilities into everyday functioning [31]. One of the promising approaches towards higher ecological validity is simulating real-life situations in virtual environments (VEs) [32]. VEs enable us to create an authentic, complex, and safe environment for assessment and training while preserving full control over the situation [32]. Furthermore, VEs can simulate so-called activities of daily living (ADLs), e.g., shopping, cooking, navigating, which are typically impaired in patients with schizophrenia [33]. The rehabilitation tasks’ resemblance to real-life conditions can itself lead to enhanced ecological validity [34] and therefore facilitate the transfer of the gained abilities into everyday functioning [35]. VEs can be presented using either immersive or non-immersive virtual reality (VR). Immersive VR technology, e.g., head-mounted displays, results in a higher sense of presence [36] which can lead to improved resemblance to real-life conditions and therefore increased ecological validity [37]. Non-immersive VR, on the other hand, is presented on a standard computer screen. In this study, the VE program was presented as a non-immersive VR.

### 1.4. AIMS

This study compares the efficacy and feasibility of the VE program with standard paper-pencil treatment in a cross-over trial.

Goals:To compare the effect of the VE program and standard treatment on cognitive outcome.To investigate the feasibility of the VE program.To evaluate the participants’ progress in the VE program.

## 2. Materials and Methods

The study was approved by the ethics committee of the NIMH CZ Ethics Committee on the 22 March 2017 under the number 105/17. All participants signed an informed consent form containing information about the experimental procedure and exclusion criteria.

### 2.1. Participants

The participants were recruited from the outpatients attending the Psychotherapeutic Day Center for psychotic patients in Karviná town. Participants were assessed by a psychiatrist according to ICD-11 standard symptoms [9] and medicated according to their diagnosis (For more details see Appendix A). A total of 35 participants were recruited. Only those participants who completed at least one program were included in the subsequent analyses: 28 in total, 17 male (M = 35.391, SD = 8.033), 11 female (M = 36.909, SD = 13.172). Twenty-two participants were diagnosed with schizophrenia or other primary psychotic disorder and six participants were diagnosed with major depressive disorder [9]. The mean duration of the illness was 6.19 (SD = 6.3) years and it ranged from 0.5 to 20 years. The level of education ranged from elementary education (*n* = 9), vocational school (*n* = 9), high school (*n* = 9) to university education (*n* = 1). 

Inclusion criteria:Diagnosis of schizophrenia or other primary psychotic disorders or diagnosis of a depressive episode or recurrent depressive disorder according to ICD-11 [9].Age 18–60 years

Exclusion criteria:Severe visual impairmentNeurological disorder or comorbid psychiatric diagnosisPhysical handicap preventing the participant from participating in the VE programRefusal to give informed consent

### 2.2. Procedure

Before the beginning of the rehabilitation program, all participants completed a baseline cognitive assessment. Participants were then randomly assigned to the first rehabilitation program: VEs program or standard treatment. Equal randomization was not possible due to the insufficient number of laptops available for the VE program. After the first program completion, participants were retested with the cognitive battery and then assigned to the second program: standard treatment or VEs program. After completing the second program, the participants were once more assessed with a cognitive battery. In some cases, participants did not complete the program due to relaps-related hospitalization or due to the adverse epidemiological situation associated with the coronavirus disease, SARS-CoV-2.

### 2.3. Virtual Environments (VEs) Program

The virtual tasks were administered on 17.3” laptops. Participants interacted with the VEs using a keyboard and a mouse. Participants attended 12 computer sessions during 6–12 weeks. Each session lasted 30 min and consisted of a set of three VEs tasks. For the VE program, we adopted a drill–practice approach and progressively increased the tasks difficulty according to the participant’s performance. During the first session, participants were explained how to interact with the virtual environment and how to control specific tasks. During the following sessions, participants were instructed to train for approximately 5 min with the Shooting gallery and then continue for 10 min with Virtual Supermarket Shopping Task and 10 min with Objects.

**Shooting gallery** is a variant of a standard go–nogo [38] task in an enriched and gamified VE. The task trains the ability to differentiate targets from non-targets, selective attention, psychomotor speed, and inhibitory control. Participants are asked to “shoot” (press a key, no aiming required) when targets (predatory animals) appear and to ignore the no-go targets (non-predatory animals).

**Objects** is a task focusing on contextual episodic memory. Participants are asked to remember the identity, location, and sequence of objects which they collect from a virtual family house. Each trial has two phases: an acquisition phase—participants are guided towards a series of objects and tasked with remembering their order and position, and a recall phase—participants are asked to select the remembered objects from a collection of various items and then instructed to place them in their original positions and the original order.

**Virtual Supermarket Shopping Task (****VSST)** is focused primarily on memory abilities. It requires participants to remember and collect items from a shopping list in a virtual supermarket (see Figure 1). Each VSST trial consisted of two phases: the acquisition phase (presentation of the shopping list) and the recall phase (shopping items collection). The recall phase followed immediately after the acquisition. The training session started with a difficulty of 2 items and the difficulty increased by one item each time the participant collected all items on the shopping list and no additional items. More detailed information about the task procedure is described in our previous work [39]. VSST has multiple performance measures: number of correctly collected items, number of extra items, trial time and trial distance.

For the purposes of this study, we only analyzed the results from VSST, as it was already validated against standard cognitive measures in patients with schizophrenia [39].

### 2.4. Standard Treatment

Participants in the standard cognitive rehabilitation program (referred to as a standard treatment) attended a total of 12 paper–pencil sessions over a period of 6–12 weeks. Each session lasted 45 min. The session started with a warm-up game and then continued with a set of paper-pencil tasks focusing on attention, fine motor skills, recall, short-term and long-term memory, verbal fluency, visual search, cognitive flexibility, abstraction and executive functions, and numerical abilities. The tasks were adapted to each participant’s own abilities.

### 2.5. Cognitive Assessment

The participants’ cognitive abilities were assessed before cognitive rehabilitation, after completing the first program, and after finishing the second program. This allowed us to precisely track their performance over long periods and evaluate the effect of each program individually.

#### The Repeatable Battery for the Assessment of Neuropsychological Status (RBANS)

The RBANS test battery was previously standardized in schizophrenia patients [40] and it covers most impaired cognitive domains in schizophrenia [10]. RBANS is easy and fast to administer, which eliminates the burden placed on participants during the assessment. Its alternative forms (A, B, C, D) allow for repeated testing after the cognitive remediation program, which is fitting for the purposes of our study. RBANS has twelve subtests that can be combined into five cognitive domains [40,41,42]: immediate memory, visuospatial/constructional, language, and attention and delayed memory.

As the purpose of this study was not a clinical assessment, but a comparison of repeated performance, we have opted to use the raw scores and not the standard metrics in RBANS related analyses. We normalized the raw scores in each RBANS domain (sum of raw scores for the corresponding subtests) for each form separately. As our models control for potential score differences between different forms, we consider this approach in our scenario to be more statistically powerful. Please note, the index scores for individual RBANS domains (instead of normalized raw scores) are presented in graphs and descriptive tables to provide representative RBANS data for clinicians.

### 2.6. Feedback Questionnaire

After the program, participants filled out a brief evaluation questionnaire. The questionnaire had 17 questions about participant’s enjoyment and perceived difficulty of the programs, their task preferences, willingness to participate in the program again, and their perception of achieved cognitive enhancement. The participants responded on a 5-point Likert scale (For details see Appendix A).

### 2.7. Adherence to the Treatment

To assess the adherence to the VE program we looked at the *refusal rate* of the participants who willingly decided to drop from the program or not attend it(therefore not due to hospitalization or adverse epidemiological situation).

### 2.8. Statistical Analysis

We chose to analyze the intervention effects on the RBANS normalized scores and the VSST improvement using the linear mixed-effects modeling approach [43]. Linear mixed-effect models are statistically more powerful than general linear models in case the data are correlated, which is the case in any study involving repeated measures. They allow modelling and removal of the effects of individual differences, such as the subjects’ cognitive aptitude or diagnosis (*random effects*), from the investigated ones (*fixed effects*). The linear mixed effect models are also able to handle unequal group sizes and missing data [44], which makes them useful in studies with psychiatric patients, where some participant dropout can be expected.

Statistical analyses were performed using R [45]. Plots were generated using the ggplot2 package [46] and linear mixed effect modeling was handled by the lme4 package [43] and *p*-values for the resulting models obtained using the package lmerTest [47].

## 3. Results

### 3.1. Adherence to the Treatment

From the initial number of 35 participants, seven did not finish any rehabilitation program, 28 completed at least one program and 17 completed the entire program (see Figure 2 for details).

Two participants refused to attend the VE program because they did not find it beneficial and one participant felt that he could not perform well in the program. This makes the total VEs program *refusal rate* 8.6%. In contrast, no participant quit or refused the standard treatment.

### 3.2. Feedback Questionnaire

The questionnaire aimed to assess participants’ subjective perception of the program in terms of difficulty, enjoyment or subjective improvement. We found no significant differences between the standard treatment and the VE program in either category (see Table 1). Needless to say, the participants seem to slightly prefer the standard treatment in terms of their willingness to repeat it in the future (*t*(48.38) = −1.91, *p* = 0.062). Generally, the participants perceived both programs as enjoyable and beneficial.

### 3.3. Virtual Supermarket Shopping Task

Using linear mixed effect models we looked at how patients learned VSST as the cognitive training progressed. We defined maximum difficulty achieved as the highest difficulty level (number of items on the list) participants had reached during the session. We observed that participants were continuously able to improve and proceed to more difficult trials as the training progressed (b = 0.09, 95% CI [0.04, 0.13], *t*(156.74) = 4.04, *p* ≤ 0.001) (see Figure 3), although the rate of improvement was small.

As the trial trajectory and trial time are dependent on the participant’s performance as well as task difficulty (more items take longer to pick up), we modeled the trial time and trajectory with both session and trial difficulty as a predictor. We observed that participants improved in both of these measures as cognitive training progressed. With each new session we saw a decrease in trial trajectories (b = −5, 95% CI [−6.84, −3.16], *t*(1268.04) = −5.33, *p* ≤ 0.001) and trial times (b = −5.61, 95% CI [−6.79, −4.42], *t*(1269.06) = −9.29, *p* ≤ 0.001).

One of the metrics we explored in our previous work and found to be indicative of cognitive performance [39] was the VSST item performance—a ratio of correctly picked items and the number of items on the list. As the item performance is dependent on task difficulty (more difficult trials lead to higher performance decline), and, as we learned in our previous work that participants only start to struggle after 5-item difficulty, we only analyzed this metric in trials beginning with a difficulty of six items. Modeling the item performance as a function of a session and task difficulty, we observed that the task difficulty significantly decreases the item performance (b = −0.06, 95% CI [−0.09, −0.03], *t*(323.9) = −3.69, *p* ≤ 0.001), but the number of completed sessions had no effect (b = 0.95% CI [0, 0.01], *t*(320.85) = 1.08, *p* = 0.281), suggesting that participants are not improving in their item collection precision. This lack of session effect could be explained by the relationship between item performance and the difficulty, which is also increasing across the sessions.

### 3.4. Cognitive Assessment

We have observed that patients suffering from schizophrenia scored lower in all RBANS domains and had significantly worse baseline performance than patients with MDDs in the language domain and RBANS total index score (see Table 2). This was one of the reasons we opted for the linear mixed effect models in our subsequent analyses as the method is able to implicitly control for the effect of the diagnosis.

Using linear mixed effect models we explored the effect of the intervention (see Figure 4) and the session on RBANS scores, with the participant being a random effect. We have not found any effect of the intervention, neither positive nor negative, on either domain (see Table 3). We only found a negative effect of the third session on the normalized total RBANS score (b = −0.31, 95% CI [−0.55, −0.06], *t*(15.41) = −2.47, *p* = 0.026) with participant’s scoring marginally worse at the end of the study (RBANS form C, both interventions completed) than after the first interventions.

Although linear effects models are suited to control for individual differences and therefore participants’ diagnoses, it is possible that the participants with affective disorders could perform differently from those with schizophrenia. We modeled the cognitive performance in each RBANS standardized domain as a function of the diagnosis, session and their interaction as fixed effects and participant as a random effect. We have not found any significant interaction between the diagnosis and the session on any RBANS domain (see Table 4).

## 4. Discussion

The primary aim of this study was to evaluate the efficacy and feasibility of a novel rehabilitation program using complex VEs for psychiatric patients. We administered the program to patients suffering from schizophrenia or MDD and compared it to standard treatment in a cross-over study. Our results show that the proposed VEs program could be, in terms of its feasibility, a suitable alternative to the standard paper-pencil approaches, which are sometimes criticized for low ecological validity [29] that can limit the treatment functional outcome [23]. Participants reported high levels of enjoyment and perceived benefit in both programs, which suggests good overall acceptance of the VE program. Nevertheless, we did not find significant improvement in any of the standardized cognitive measures. The impact of the rehabilitation on participant’s cognitive improvement did not significantly differ between the programs. But the possibility of tracking participants’ continuous progress in the VE program allowed us to report their significant improvement in the VEs task over the program duration.

### 4.1. Cognitive Assessment

We measured participants’ standard cognitive performance in the beginning and then after each program completion. The VE program did not differ from the standard treatment in terms of its cognitive outcome and we did not find any effect of either program on cognitive measures. Our results only indicate worse performance in the last assessment session in comparison with the second session in general (independent of the applied order of training programs, see Limitations for more details). These findings are in contrast to the previous findings showing a positive impact of the VR program on cognitive functions in patients with schizophrenia when using immersive technology [48,49,50].

Although previous meta-analyses revealed significant improvement in different cognitive domains in schizophrenia after cognitive remediation [24,25,26] the meta-analysis by Revell et al. [51] investigating CR efficacy in early schizophrenia showed a significant effect on verbal memory and learning, but only a non-significant small effect of CR on global cognition and other cognitive domains. Similarly, some randomized clinical trials also failed to report significant improvement [52,53,54], arguing the potential inefficacy of the cognitive remediation approach in schizophrenia patients in general [53]

Both treatments in this study used a drill–practice approach. A recent meta-analysis [55] emphasizes the benefit of strategy-focused CR over drill practice in the rehabilitation of verbal and visual learning. In future studies, it could be beneficial to incorporate strategy learning into the VE program as the deficit in the mnemonic strategies was proposed as a core feature of the memory deficit in schizophrenia [56,57]. Moreover, according to Lejeune et al. [55] discussing how to apply acquired skills in real-life, so-called “bridging”, is a beneficial approach for global cognition and verbal memory. Combining VEs programs with “bridging” and strategy learning could be therefore more effective as the complex VEs provide a continuous path between the training and real-life skills.

Furthermore, it is possible that our sample was specific due to sociodemographic characteristics and prevalence of clinical symptoms (e.g., negative symptomatology, lack of insight) that can interfere with CR outcomes (for review see [58]). As we did not measure clinical symptoms during the intervention and our assumption is based merely on observation of the present therapist, we cannot draw any conclusions.

In this study, we focused on two main diagnostic categories—schizophrenia and other psychotic disorders and depressive disorders [9]. Even though these diagnoses show similar patterns of cognitive deficits with more profound impairment in schizophrenia, the course of the cognitive deficit and response to the CR can differ [6,7]. We did observe that participants diagnosed with schizophrenia scored lower in the cognitive tests, which is consistent with the literature [3,6]. Nevertheless, we did not find any significant interaction between the diagnosis and the session on the RBANS scores, suggesting that treatment affects both groups similarly.

Another explanation for not finding the expected beneficial effects of CR can be the low sensitivity of the RBANS battery to detect the improvement, which, especially in the memory domain, can be subtle [22,24,55]. It could be useful to apply a more targeted battery, e.g., MATRICS [41,59].

Last but not least, the significant effect of cognitive remediation reported by meta-analyses [23,24,26] could be overstated due to a publication bias [60]. For example, a recent meta-analysis by Cella et al. [26] calculated publication-risk and pointed out a high risk in the domains of attention and vigilance, and executive functioning.

### 4.2. VEs Program Performance

The VE program enables clinicians to automatically monitor and record participant’s performance and adjusts difficulty according to participant’s performance. This is a major advantage over the standard paper–pencil programs, where it would be time-consuming or even impossible to achieve. The data collected in the VE program allow for immediate or retroactive automated analysis and aim to achieve a more objective assessment of a patient’s day-to-day improvement or stagnation. This is not only beneficial to the clinician but can also provide useful feedback to the participant throughout the CR at no additional time cost.

According to our results, the participants improved in VSST as the program progressed. We found a continuous improvement in the maximum achieved difficulty, meaning that participants were able to make fewer mistakes and memorize more shopping items in later sessions. Nevertheless, the participants did not improve in the item performance measure—a ratio of correctly picked items. This can be due to the fact that the VSST had an automatized increasing difficulty (increasing number of items on the shopping list) constantly challenging the participant and resulting in stable achievement in the item performance measure. Besides their recall performance, the participants were also able to solve the task faster and use shorter trajectories which shows increased efficiency with the task.

### 4.3. VEs Program Feasibility

We recruited a total of 35 participants, out of whom 28 completed at least one program and 17 finished both. Three participants out of 35 refused the VE program resulting in an 8.6% refusal rate, while no participants refused the standard treatment.

This difference in refusal rate between the two training programs can be explained as follows. During the standard treatment, participant’s performance is not evaluated and no pressure is put on patients to actively participate in the training. Also, the standard treatment is administered in groups and offers more social interaction, it includes warm-up games or board games and is generally more relaxed. This could explain why some participants rejected the VE program that, in contrast, was administered individually and required the participants to actively contribute.

In our sample, two participants refused the VE program after a short demonstration and claimed they did not like it, and one participant was not willing to participate because of the fear of failure. Interestingly, only participants with schizophrenia refused to attend the VE program, so the reluctance could be related to schizophrenia’s negative symptomatology, e.g., abulia and apathy [61,62].

According to the feedback questionnaire, participants who attended the programs perceived them similarly in terms of benefit, difficulty, enjoyment and perceived cognitive enhancement. The only difference we have found was a trend towards higher willingness to repeat the standard treatment (*t*(48.38) = −1.91, *p* = 0.062). So, participants who completed the VE program did not dislike it, nor refused to participate again in the future. In general, participants rated both programs as enjoyable and beneficial.

### 4.4. Limitations

The unequal sizes of experimental groups are a major limitation of this study. This issue should be to some extent addressed by the cross-over study design and the implemented statistical models, but some bias may remain. The technical equipment of the Day Center (e.g., limited number of laptops) did not allow for full randomization, and an adverse epidemiological situation led to a significant drop-out rate. The second limitation is with the RBANS battery and its forms. Although the RBANS forms are comparable in terms of difficulty [63,64], our data shows some variability in the ‘language’ domain. As the forms were administered in the same order to all participants (baseline was tested with form A, form B was administered after the first program and form C after the second program), it complicates the separation of the session effect from the effect of the RBANS form itself. Although the linear mixed effect models and our own RBANS normalization can, to a large extent, account for these imperfections, some of the effects we would have expected to see, such as general improvement over time, can remain confounded. In future studies, the randomization of the RBANS forms or application of different cognitive measures is recommended.

Thirdly participants filled in the feedback questionnaire after completing the entire study, which might have affected their ability to differentiate between the two programs.

Moreover, the functional outcome of patients was not evaluated. In future studies, it would be beneficial to measure the quality of life or everyday functioning throughout the program, as it was proposed that complex VEs can improve the transfer of the learned abilities to real-life [31].

## 5. Conclusions

The results of this study indicate that neither the standard cognitive remediation or the proposed VE program resulted in improvement in standard cognitive measures.

Nevertheless, we suggest that the proposed VE program could present a suitable alternative to the standard remediation program in patients with schizophrenia and MDD in terms of its feasibility. In comparison to the standard treatment, the VE program allows easier administration with minimal effort from the clinician or therapist and automatically records the patient’s performance. This feature allowed us to evaluate the participants’ performance in the trained virtual task that improved in the course of training in terms of maximum achieved difficulty and general efficiency.

Considering the adherence towards treatment, participants perceived both programs as beneficial and enjoyable but clinicians should take into account that the patients with schizophrenia can be more prone to drop out from the treatment when facing higher cognitive demands or to be more reluctant to participate in an unfamiliar treatment.

## Figures and Tables

**Figure 1 ijerph-18-09081-f001:**
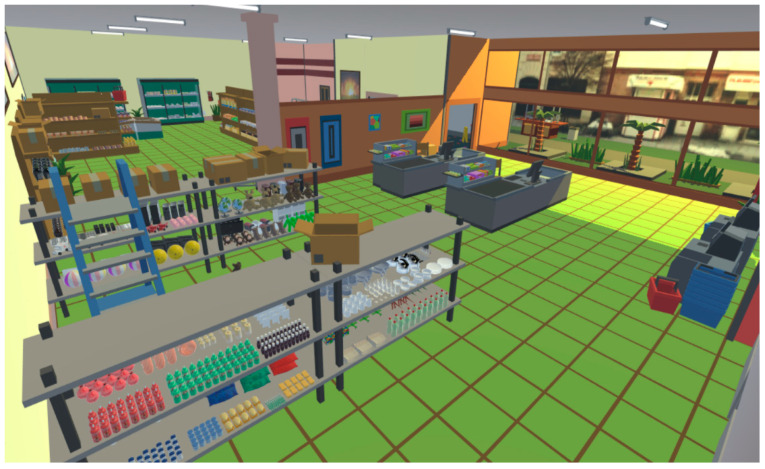
Virtual supermarket shopping task (VSST). Overview of the virtual supermarket layout.

**Figure 2 ijerph-18-09081-f002:**
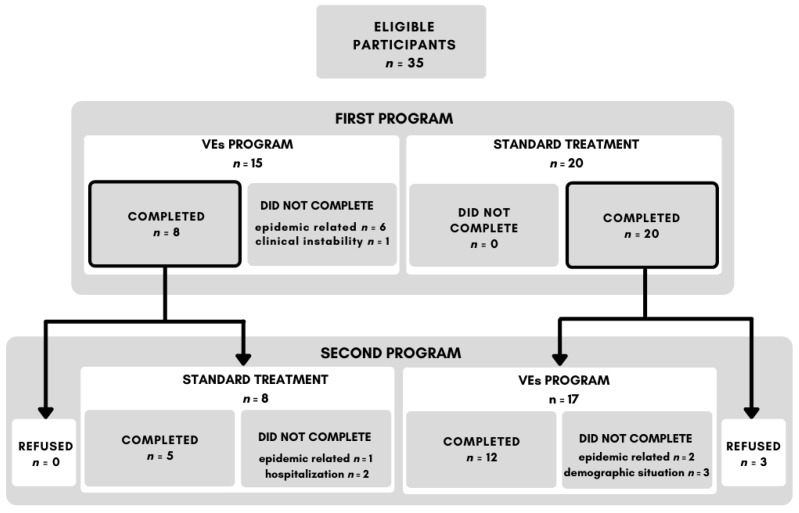
Adherence to the treatment and study procedure flow chart.

**Figure 3 ijerph-18-09081-f003:**
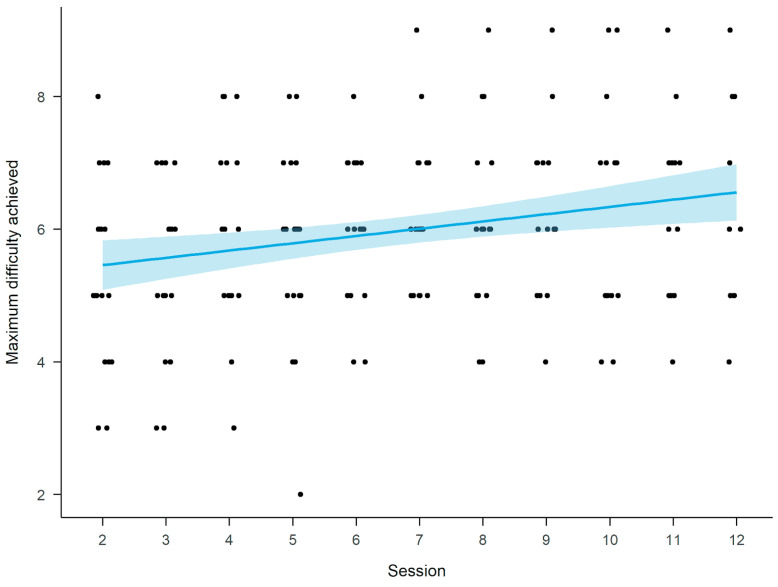
Maximum difficulty achieved in Virtual Supermarket Shopping Task plotted as an effect of the session.

**Figure 4 ijerph-18-09081-f004:**
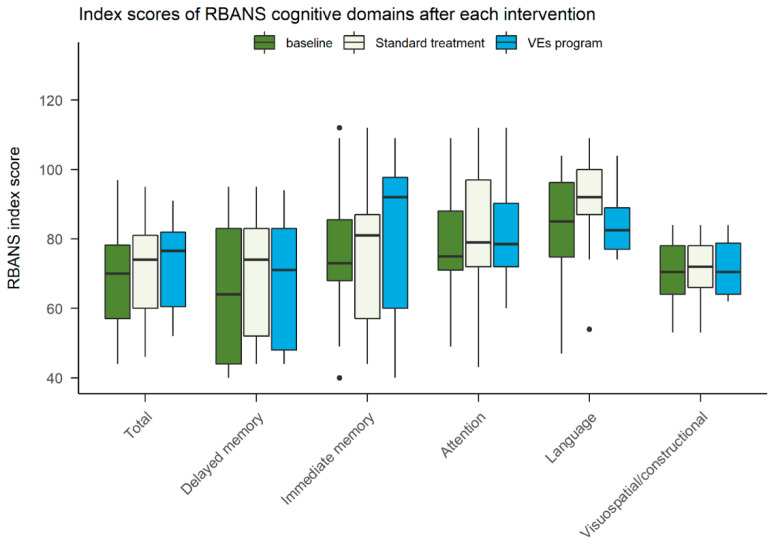
Summative results for RBANS cognitive domains (index scores) for each treatment. Boxplots represent the following information: the line is plotted at the median, the box extends from 25th to 75th percentiles, the whiskers are drawn up/down to the 10th and 90th percentile, and points represent the outliers. Legend: RBANS—Repeatable Battery for the Assessment of Neuropsychological Status.

**Table 1 ijerph-18-09081-t001:** Feedback questionnaire results for each treatment and comparison using paired *t*-test.

Question Area	Standard Treatment (*n* =27)	VEs Program (*n* = 24)	*t*-Test
perceived benefit	4.259(0.656)	4.083(0.654)	*t*(48.34) = 0.96, *p* = 0.343
perceived difficulty	2.889(0.934)	2.917(0.929)	*t*(48.36) = −0.11, *p* = 0.916
Enjoyment	4.259(0.712)	4.042(1.042)	*t*(39.98) = 0.86, *p* = 0.395
subjective improvement	3.556(0.934)	3.542(0.884)	*t*(48.79) = 0.05, *p* = 0.957
willingness to repeat	3.889(1.121)	3.231(1.366)	*t*(48.38) = 1.91, *p* = 0.062

**Table 2 ijerph-18-09081-t002:** Baseline results of RBANS cognitive domains index scores presented separately for both diagnoses and group comparison using an independent *t*-test.

RBANS Domain	Depressive Disorder (*n* = 6)	Schizophrenia (*n* = 22)	*t*-Test
	Mean(SD)		
delayed memory	77(19.411)	61.909(18.296)	*t*(23.42) = 1.50, *p* = 0.147
immediate memory	85.667(18.129)	72.5(17.88)	*t*(23.27) = 1.65, *p* = 0.112
attention	86.5(14.293)	78.273(18.224)	*t*(24.43) = 1.95, *p* = 0.063
language	93.5(6.504)	82.182(14.634)	*t*(35.27) = 2.88, *p* = 0.007
visuospatial/constructional	73.667(7.501)	70.409(9.854)	*t*(26.42) = 1.51, *p* = 0.144
RBANS TOTAL INDEX SCORE	78.5(11.467)	66(13.238)	*t*(24.09) = 2.51, *p* = 0.019

Legend: RBANS—Repeatable Battery for the Assessment of Neuropsychological Status.

**Table 3 ijerph-18-09081-t003:** The effect of the intervention (standard treatment/VEs program) on cognitive performance. The beta signifies the effect of the VE program on normalized score in the given domain while controlling for the session and participant effects.

RBANS Standardized Domain	Linear Mixed-Effect Model Result of the Virtual Environments Program in Comparison to Standard Treatment		
	Beta	95% Confidence Interval	Statistic
immediate memory	0.19	−0.1, 0.49	*t*(16.56) = 1.27, *p* = 0.221
delayed memory	−0.14	−0.34, 0.07	*t*(15.89) = −1.3, *p* = 0.212
attention	−0.03	−0.4, 0.33	*t*(16.16) = −0.18, *p* = 0.862
language	0.25	−0.21, 0.71	*t*(17.07) = 1.06, *p* = 0.302
visuospatial/constructional	−0.37	−0.93, 0.19	*t*(20.09) = −1.29, *p* = 0.210
RBANS TOTAL SCORE	0.02	−0.22, 0.26	*t*(15.7) = 0.16, *p* = 0.875

Legend: RBANS—Repeatable Battery for the Assessment of Neuropsychological Status.

**Table 4 ijerph-18-09081-t004:** Interaction between the diagnosis and session on cognitive performance. The beta signifies the interaction effect of schizophrenia and session in each RBANS domain while controlling for the participant effect.

RBANS Standardized Domain	Beta	95% Confidence Interval	Statistic
immediate memory	0.19	−0.16, 0.54	*t*(43.78) = 1.08, *p* = 0.285
delayed memory	0.21	−0.08, 0.5	*t*(43.69) = 1.44, *p* = 0.157
attention	0.07	−0.34, 0.49	*t*(43.94) = 0.34, *p* = 0.732
language	−0.18	−0.72, 0.35	*t*(44.39) = −0.68, *p* = 0.502
visuospatial/constructional	−0.1	−0.56, 0.36	*t*(43.94) = −0.42, *p* = 0.675
RBANS TOTAL SCORE	0.13	−0.14, 0.4	*t*(43.4) = 0.95, *p* = 0.346

Legend: RBANS—Repeatable Battery for the Assessment of Neuropsychological Status.

## Data Availability

The data used for this study is available upon request.

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
