# Peer review of "Cognitive Remediation in Virtual Environments for Patients with Schizophrenia and Major Depressive Disorder: A Feasibility Study"

_ijerph, 2021, doi:10.3390/ijerph18179081_

Round 1
Reviewer 1 Report
Title: Cognitive remediation in virtual environments for patients with schizophrenia and major depressive disorder: a feasibility study
Summary:
The authors have evaluated the feasibility of a computerized intervention in patients with schizophrenia and major depressive disorder, comparing it with standard paper-pencil strategies. They assessed the performance of participants in a Virtual Supermarket Shopping Task (VSST) and the RBANS. Participants perceived computerized and traditional rehabilitation strategies as enjoyable and beneficial. VSST training produces an increase of the maximum difficulty achieved by the participants and a decrease in trial trajectories and trial times. However, there is not improvement in the item performance measure. Any of the interventions showed effect for RBANS scores.
Broad comments
The topic of this work is of interest. The hypothesis of work is reasonable and justified. Procedures and results are appropriately described. Nevertheless, I have some concerns:
Abstract:
The authors state that “Participant’s performance in the VE task significantly improved in terms of memory performance (p=<0.001)”. However, such statement should clearly refer to the increase in the maximum difficulty achieved by the participants. And point out the lack of effect on the item performance (which they found to be indicative of cognitive performance in a previous work).
The authors should mention that any intervention improved cognitive measures according to RBANS scores.
Introduction:
The authors should differentiate between non-immersive and immersive virtual reality software, addressing the different level of immersion and its implication for rehabilitation, particularly the resemblance to real life and ecological validity. The computerized strategy used by the authors should be defined as non-immersive VE. They could also cite a recent review by Bisso et al. (2020) [10.3390/ijerph17176111] and the cited papers by La Paglia et al.
Discussion:
The authors state that the participants improved in the VSST, particularly the maximum achieved difficulty level. However, there is not any effect for item performance measures or the RBANS scores. This should discussed stating that there is not evidence showing that this intervention improved memory or any other cognitive domain in the participants. The same for standard intervention. Therefore, even though VE is shown to be a feasible procedure, and participants are happy with the task, the present work does not prove the usefulness of VE to improve cognition in these patients. This is the most important effect of the work and should be discussed first.
Conclusion:
The proposed VE cannot be considered a suitable alternative to the standard remediation as it has not shown to be effective.
Minor concerns:
Line 91. Include the reference number for the approval by the ethics committee.
Please, improve the quality of figure 2. It is difficult to read.
Author Response
General response: We would like to thank the reviewer for the valuable comments. We believe that the revised manuscript improved thanks to the reviewer’s input. We reworked the manuscript in order to make our findings more clear and unambiguous and we considered all reviewer’s suggestions. Moreover, during the review process, we noticed one of the participants was incorrectly categorized as a schizophrenia patient. Due to this, the results in table 2 containing the baseline measures changed. We apologize for the mistake.
Broad comments
The topic of this work is of interest. The hypothesis of work is reasonable and justified. Procedures and results are appropriately described. Nevertheless, I have some concerns:
Abstract: The authors state that “Participant’s performance in the VE task significantly improved in terms of memory performance (p=<0.001)”. However, such a statement should clearly refer to the increase in the maximum difficulty achieved by the participants. And point out the lack of effect on the item performance (which they found to be indicative of cognitive performance in previous work). The authors should mention that any intervention improved cognitive measures according to RBANS scores.
Response 1: We thank the reviewer for the valuable comment. We specified the main findings in the abstract according to the reviewer’s suggestion. We want to mention that the methodology in our previous work, where we found an increase in item performance, differed from the current methodology. In the previous manuscript (Plechatá et al., 2021) the number of tasks remained the same - each participant was only presented with 4 trials at difficulties 3,5,7,9 items. In the current study, the number of items differed from session to session. If participants did well, the difficulty increased. This meant that as the item performance increased (participants have to achieve a perfect score to progress), the difficulty increased, which in turn drove the item performance lower. This eventually led to patients finding a certain equilibrium over time. The item performance is a good metric in case participants are only scored on a predefined number of tasks, but does not fully reflect the cognitive improvements they might achieve over time in tasks with progressive difficulty.
Introduction:
The authors should differentiate between non-immersive and immersive virtual reality software, addressing the different level of immersion and its implication for rehabilitation, particularly the resemblance to real life and ecological validity. The computerized strategy used by the authors should be defined as non-immersive VE.
Response 2: We thank the reviewer for the valuable comment. We specified that the applied rehabilitation program was non-immersive, see lines 85-89.
They could also cite a recent review by Bisso et al. (2020) [10.3390/ijerph17176111] and the cited papers by La Paglia et al.
Response 3: We thank the reviewer for a relevant suggestion. We cite Bisso et al (2020) on line 88 and paper from La Paglia et al. in Discussion section on line 339.
Discussion:
The authors state that the participants improved in the VSST, particularly the maximum achieved difficulty level. However, there is not any effect for item performance measures or the RBANS scores. This should discussed stating that there is not evidence showing that this intervention improved memory or any other cognitive domain in the participants. The same for standard intervention. Therefore, even though VE is shown to be a feasible procedure, and participants are happy with the task, the present work does not prove the usefulness of VE to improve cognition in these patients. This is the most important effect of the work and should be discussed first.
Response 4: We thank the reviewer for the relevant comment. We reorganized the discussion in order to affirm the most important results. Moreover, we emphasized our findings that neither the VEs program nor standard intervention led to improvement in cognitive performance (see line 325).
Conclusion:
The proposed VE cannot be considered a suitable alternative to the standard remediation as it has not shown to be effective.
Response 5: We thank the reviewer for the important comment. We reworked the study conclusions to present the findings in an unambiguous way (lines 483-492).
“The results of this study indicate that neither the standard cognitive remediation or the proposed VEs program resulted in improvement in standard cognitive measures. Nevertheless we suggest that the proposed VEs program could present a suitable alternative to the standard remediation program in patients with schizophrenia and MDD in terms of its feasibility. In comparison to the standard treatment, the VEs program allows easier administration with minimal effort from the clinician or therapist and automatically records the patient’s performance. This approach allowed us to evaluate the participants’ performance in the trained virtual task that improved in the course of training in terms of maximum achieved difficulty and general efficiency. “
Minor concerns:
Line 91. Include the reference number for the approval by the ethics committee.
Response 6: We thank the reviewer for noticing the omission of the reference number. We have included it in the revised MS.
Please, improve the quality of figure 2. It is difficult to read.
Response 7: We thank the reviewer for the comment. We enlarged the font in Figure 2 to make the figure more readable. The figure is on line 232 of the revised MS.
Reviewer 2 Report
The present work focuses on the feasibility and efficacy of a novel rehabilitation program of Cognitive Remediation in patients with psychiatric disorders. Specifically, the authors describe a complex virtual environments (VEs) with standard paper-pencil 14 treatment in patients with schizophrenia and major depressive disorder.
The available literature describes cognitive impairments in various psychiatric disorders such as schizophrenia and mood disorders.
Cognitive impairments in schizophrenia are widespread in
most cognitive domains, and declines in attention, processing
speed, memory, executive function, language, and social
cognitive function can be observed.
People with major depressive disorder experience
a decline in cognitive functions such as attention, learning
and memory, processing speed, and executive function.
Declined cognitive function is known to predict non-response to
treatment of depressive disorder and functional impairment, and
is related to lower quality of life. Moreover, it has been
suggested that cognitive function continues to decline, not only
during the depressive episode, but also in euthymic states (see Kim et al., 2018).
While for schizophrenia several meta-analyses on Cognitive Remediation effectiveness have been published to date, there are only a few studies addressing Cognitive Remediation in major depressive disorder patients.
One first observation is that the authors state that the patients were recruited from the Day Center for psychotic patients. The question is: the patients diagnosed with major depression were not referred to this facility, or they were but were found to be not psychotic but depressed?
In the present study, since the authors focused on two main diagnoses that can differ in the cognitive profile, it is not clear if the two populations had differences at the baseline cognitive assessment.
Moreover, the authors state that the linear mixed-effects model controls for individual differences, including the effect of the diagnosis. Nevertheless, since so poor literature exists on major depression and Cognitive Remediation, it would be nicer to have results presented as divided for diagnosis.
Author Response
General response: We would like to thank the reviewer for the valuable comments. We believe that the revised manuscript improved thanks to the reviewer’s input. We included analyses investigating the differences between the two diagnostic groups and discussed the results in the relevant section. Please see specific comments to your questions below.
One first observation is that the authors state that the patients were recruited from the Day Center for psychotic patients. The question is: the patients diagnosed with major depression were not referred to this facility, or they were but were found to be not psychotic but depressed?
Response 1: The name of the Psychotherapeutic Day Center does not completely correspond to the diverse diagnostic groups that are referred to this facility. Even though the majority of the patients are diagnosed with schizophrenia or other primary psychotic disorder, there are also patients with mood disorders without psychotic symptoms attending the program.
In the present study, since the authors focused on two main diagnoses that can differ in the cognitive profile, it is not clear if the two populations had differences at the baseline cognitive assessment.
Response 2: We thank the reviewer for the question. Please notice Table 2 on line 281 showing the group baseline differences and relevant statistics. We found significantly lower performance in schizophrenia patients for overall RBANS score and language domain. During the review process, we noticed that one participant was incorrectly categorized as a schizophrenia patient, thus the results in Table 2 were corrected, with no change to the overall results. We apologize for the mistake.
Moreover, the authors state that the linear mixed-effects model controls for individual differences, including the effect of the diagnosis. Nevertheless, since so poor literature exists on major depression and Cognitive Remediation, it would be nicer to have results presented as divided for diagnosis.
Response 3: We thank the reviewer for the valuable comment. As there is indeed not enough evidence to support that the response on CR in both diagnostic categories is similar, we modeled the RBANS cognitive outcome as a function of both session and diagnosis and their interaction. We have not found any significant effects of the interaction, suggesting that patients with either diagnosis were affected by the treatment in the same way. The other results mirrored those reported previously. In these models, schizophrenia diagnosis had an overall negative effect on RBANS scores, although we did not find it significantly different from the MDD, see table below.
Suppl. Table. 1 Effect of diagnosis (schizophrenia diagnosis in contrast to MDD) in the linear mixed effect model predicting individual RBANS domains as a function of diagnosis, session, interaction (diagnosis*session) as fixed effects and participants as a random effect.
RBANS TOTAL SCORE |
b=−0.59, 95% CI [-1.6, 0.42], t(42.09)=−1.14, p=.259 |
Immediate memory |
b=−0.61, 95% CI [-1.67, 0.46], t(52.88)=−1.12, p=.269 |
Delayed memory |
b=−0.4, 95% CI [-1.41, 0.61], t(45.5)=−0.78, p=.440 |
Attention |
b=−0.53, 95% CI [-1.65, 0.59], t(61.55)=−0.93, p=.357 |
Language |
b=−0.36, 95% CI [-1.58, 0.86], t(68.92)=−0.58, p=.562 |
Visuospatial/Constructional |
b=−0.01, 95% CI [-1.18, 1.16], t(65.19)=−0.02, p=.985 |
We have also modeled each RBANS domain normalized score by fitting a session as a fixed effect and participants as a random effect (normalized rbans dimension ~ session + (1|participant)) separately for patients with MDD and patients with schizophrenia. Neither group significantly changed in their RBANS performance with progression sessions.
Suppl. Table. 2 Session as a predictor of RBANS performance for patients with affective disorders. Coefficients are for the session as a linear predictor (1,2,3)
RBANS TOTAL SCORE |
b=−0.23, 95% CI [-0.51, 0.05], t(9.24)=−1.61, p=.141 |
Immediate memory |
b=−0.25, 95% CI [-0.67, 0.18], t(9.65)=−1.14, p=.281 |
Delayed memory |
b=−0.24, 95% CI [-0.53, 0.05], t(9.15)=−1.63, p=.137 |
Attention |
b=−0.18, 95% CI [-0.45, 0.09], t(9.23)=−1.29, p=.230 |
Language |
b=0.08, 95% CI [-0.39, 0.55], t(14)=0.33, p=.746 |
Visuospatial/Constructional |
b=0, 95% CI [-0.29, 0.28], t(9.23)=−0.03, p=.980 |
Suppl. Table. 3 Session as a predictor of RBANS performance for patients with schizophrenia. Coefficients are for session as a linear predictor (1,2,3)
RBANS TOTAL SCORE |
b=−0.1, 95% CI [-0.22, 0.02], t(34.26)=−1.66, p=.106 |
Immediate memory |
b=−0.06, 95% CI [-0.21, 0.08], t(34.43)=−0.85, p=.401 |
Delayed memory |
b=−0.03, 95% CI [-0.16, 0.11], t(34.58)=−0.4, p=.691 |
Attention |
b=−0.1, 95% CI [-0.31, 0.11], t(34.85)=−0.94, p=.353 |
Language |
−0.12, 95% CI [-0.37, 0.13], t(35.15)=−0.91, p=.369 |
Visuospatial/Constructional |
b=−0.09, 95% CI [-0.33, 0.14], t(34.81)=−0.78, p=.443 |
As these models convey the same information as the interaction model, we only included the interaction model in the manuscript. We added the following paragraph to the manuscript (lines 299-307) and added Table 4 (page 10).
“Although the linear effects models are suited to control for individual differences and therefore participants’ diagnoses, it is possible that the participants with affective disorders could perform differently from those with schizophrenia. We modeled the cognitive performance in each RBANS standardized domain as a function of the diagnosis, session and their interaction as fixed effects and participant as a random effect. We have not found any significant interaction between the diagnosis and the session on any RBANS domain (see Table 4).”
Table 4 Interaction between the diagnosis and session on cognitive performance. The beta signifies the interaction effect of schizophrenia and session in each RBANS domain while controlling for the participant effect.
RBANS standardized domain |
Beta |
95% Confidence interval |
Statistic |
Immediate memory |
0.19 |
-0.16, 0.54 |
t(43.78)=1.08, p=.285 |
Delayed memory |
0.21 |
-0.08, 0.5 |
t(43.69)=1.44, p=.157 |
Attention |
0.07 |
-0.34, 0.49 |
t(43.94)=0.34, p=.732 |
Language |
-0.18 |
-0.72, 0.35 |
t(44.39)=−0.68 , p=.502 |
Visuospatial/Constructional |
-0.1 |
-0.56, 0.36 |
t(43.94)=−0.42, p=.675 |
RBANS TOTAL SCORE |
0.13 |
-0.14, 0.4 |
t(43.4)=0.95 , p=.346 |
Moreover, we added the following lines (360-364) to the Discussion section:
“We did observe that participants diagnosed with schizophrenia scored lower in the cognitive tests, which is consistent with the literature [3,6]. Nevertheless, we did not find any significant interaction between the diagnosis and the session on the RBANS scores, suggesting that treatment affects both groups similarly. “
Round 2
Reviewer 1 Report
Thanks for the reply and the changes made in the manuscript. It has been improved and can be published now.
Author Response
We thank the reviewer for the valuable input during the peer-review process.